# CTCR Prototype Development:
# An Obstacle in the Research Community?

Reinhard M. Grassmann[*†], Sven Lilge[*], Phuong H. U. Le[‡], Jessica Burgner-Kahrs[*]

[*] Department of Mathematical and Computational Sciences
University of Toronto Mississauga, Toronto, ON M4P 1A6, Canada
[†] Email: reinhard.grassmann@utoronto.ca
[‡] Department of Computer Science
University of Toronto, Toronto, ON M5S 1A1, Canada

*Abstract*—Due to their small size and ability to follow non-linear paths, concentric tube continuum robots might offer new opportunities for minimally invasive surgery. Currently, several research groups in this relatively young field are working on early-stage prototypes to develop technologies and methodologies to realize this envisioned application. To date, a large variety of different prototypes exists across the individual research groups.

In this paper, current habits of the research community are investigated that cause such a proliferation of many different prototypes. System thinking tools are applied to provide an in-depth analysis of the dynamics between publication pressure and developing prototypes. According to a gathered data set based on 139 publications in this research field, 53.2 % of the publications include a robot-based evaluation. However, 61.1 % of 36 prototypes are only used in one publication, indicating a low prototype reuse rate. As a result, a lot of time is devoted to create prototypes which hinders the development of a robotic platform. Such a robotic platform can leverage the whole research community by focusing efforts on the main research challenges.

## I. INTRODUCTION

Concentric tube continuum robots (CTCR), shown in Fig. 1, are a new class of continuum robots proposed and introduced simultaneously by Webster et al. [48] and by Sears and Dupont [42] in 2006. Since then, numerous prototypes and new approaches have been proposed and studied. It can be observed that progress made by the research community also fosters progress in steerable needles and, according to Walker et al. [47], also in hyper-redundant snake-like robots. In addition, two surveys by Gilbert et al. [16] and by Mahoney et al. [35] exclusively focusing on CTCR have been published in 2016. However, even after nearly 15 years of continued growing research and progress, neither a commercial product nor an accessible robotic platform are available. We note, that there are efforts in the form of startups, e.g. Virtuoso Surgical Inc. founded around Webster, to commercialize CTCR.

*a) State of CTCR Research:* Gilbert et al. [16] provide a comprehensive description of the state of the art as well as a thorough discussion on the development history. They mention that the field is maturing and that one can hardly find a medical robotics forum that does not include several presentations on CTCR. By analyzing broader trends throughout the research community, Gilbert et al. [16] propose future directions and thoroughly discuss open opportunities. As an extension of the review by Gilbert et al. [16], a followup review on CTCR by

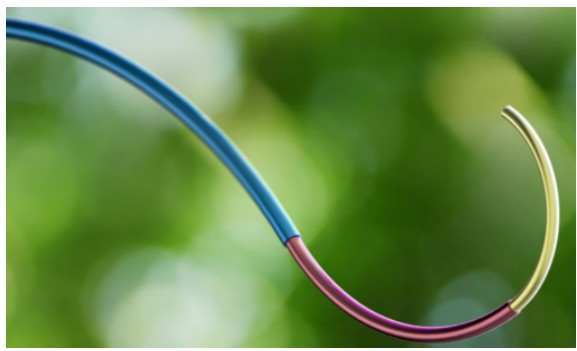

Fig. 1: Rendering of a CTCR (Image from [8]). A CTCR consists of several super-elastic tubes. These are the one of the smallest robot among all continuum robots [16, 35]. According to Burgner-Kahrs et al. [5], these have an ideal structure for the usage in robot-assisted minimally invasive surgery.

Mahoney et al. [35] includes advancements in actuator and tube design, control and modeling, as well as in planning and sensing.

Despite many suggested medical and inspection applications, few have been studied in depth. One promising application is the usage in robot-assisted minimally invasive surgery. A rigorous review of the application of CTCR in medicine is provided by Burgner-Kahrs et al. [5], which reflects on the trend and consensus of CTCR in their envisioned applications.

The three mentioned reviews envision a bright CTCR future by continuing the recent advancements in the research community. While we share this sentiment, we also believe that some of the current practices and habits of the research community are hindering the progress towards a widely used common robotic platform. Currently, due to the lack of such a platform, many individual prototypes exist. However, a robotic platform would offer easy access to existing CTCR technologies and allow researchers to tackle the remaining major challenges in control, design, modeling and sensing. These challenges are fundamentally coupled [36]. On top of that, most well-known modeling techniques used for serial kinematic manipulator are not applicable for CTCR.

*b) Contribution:* In the present work, we show that current research habit promote the use of prototypes over the development of a robotic platform and, therefore, are limiting

the progress towards one. We provide a systematic analysis of this habit using systems thinking tools accompanied by an extensive data set on CTCR publications. In particular, the contributions of this paper include:

- A causal loop diagram based description of an interdependency of publication pressure and a variety of prototypes.
- A data-driven confirmation based on a manually gathered and augmented data consisting of CTCR publications.
- As a minor contribution, a proposed metric for quantifying the influence of a prototype, which is straightforward to adapt to methods and approaches.

## II. AN EMPIRICAL OBSERVATION

### A. Research Field of Serial Kinematic Manipulator

A robotic revolution can be observed in the research field of serial kinematic manipulators, the most commonly known industrial robots. In the following, we will highlight three of the most recognizable robots, although several other robots contribute to the development described below as well.

*a) PUMA 560:* The robot is depicted in Fig. 2a. Since its release in 1979, the PUMA 560 has become one of the major references in robotics research due to its versatility and usability [23]. As a result, it is well studied and its parameter are very well known. According to Corke [9], this robot is often referred to as the *white rat* of robotics research. In homage to its important role in robotic community, the PUMA 560 is still widely used in across many robotic textbooks.

*b) DLR LWR III:* In comparison to other industrial robots, the DLR LWR III developed by Hirzinger et al. [24] is compliant, sensitive, and ensures safety in the sense that intended or unintended physical interaction with human and environment can be detected and reacted to, see Fig. 2b and Fig. 2c. Those innovative features are an emergent property of being redundant and able to measure forces.

This robot is considered as the enabler for safe physical human-robot interaction [21] which is a new domain gained significant economic relevance recently, see Fig. 2. Since its commercialization by the Manufacturer Kuka AG, industry and research labs have access to the commercialized version of the new robot generation. However, this accessibly is restricted due to the high cost of about 100.000 US$.

*c) PANDA:* Due to the physical human-robot interaction shown in Fig. 2d, real-world applications as well as research goals are currently experiencing a fundamental paradigm shift [21]. However, research with the torque-controlled lightweight robots, that are required for this particular task, has long been limited to highly sophisticated and expensive robots such as the DLR LWR III. The PANDA, revealed to the public in 2017, is a state-of-the-art yet affordable industrial robot with an easy-to-use software. The robot is easily accessible even to non-experts, as a pilot project [22] for students has shown. The PANDA costs a tenth of a similar high performance robot, making it accessible even for smaller organizations and research labs. As such, it is becoming an increasingly common robot in the robotics community, which may lead to a *sensitive white rat* of robotic research.

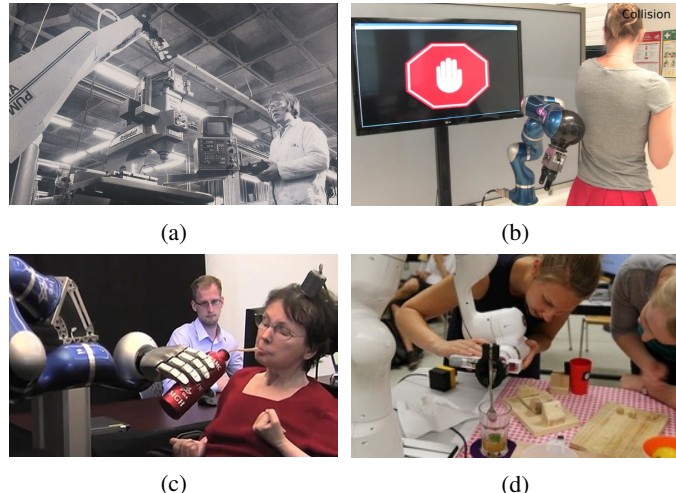

Fig. 2: Paradigm shift in the research of serial kinematic manipulator. (a) PUMA a common robot in the early robotic research. (b) Unintended physical interaction. (Image form [17]) (c) Intended physical interaction. (Image form [25]) (d) PANDA also known as FRANKA in a physical human-robot interaction scenario during a *Robothon* (Image ©roboterfabrik, by courtesy by Institute of Automatic Control, Leibniz University Hannover).

### B. Missing white rat of CTCR Research

Considering the advances in software and hardware in overall robotics, we pose the question of why this rate of advancement has not been mirrored in the our research community. The research community should be poised to create the next generation of CTCR, extending the scope of development to increasingly complex applications, such as advanced surgical scenarios. Additionally, we draw the reader's attention to the the lack of a robotic platform, which would accelerate progress for the research community as the previously discussed robots did. Therefore, we raise the question - where is the *white rat* of CTCR research?

It is our hypothesis that a large and growing number of different existing prototypes, which is the result of current practices and behaviours in our research community, imposes a barrier for the development of an accessible robotic platform. Throughout this paper, we define a robotic platform as a well-developed physical CTCR including software, that provides fundamental methods like control algorithms and kinematic modeling. It is well documented, easily accessible and usable by individual research groups in a straightforward manner. Thus, a robotic platform differs largely from early-stage robotic prototypes.

## III. METHODS

In this section, we give a short introduction to CTCR. Further, our data acquisition is stated and the data set is introduced. Afterwards, we propose a metric to analyse the sharing of prototypes. Lastly, we briefly describe the concept of causal loop diagrams.

TABLE I: Key words used for the data acquisition. As this table suggests, the name of this type of robot is not used consistently in the literature.

| Abbreviation | Used synonym |
|---|---|
| AC | Acitve Cannula |
| CMTCatheter | Curved Multi-Tube Catheter |
| CMTD | Curved Multi-Tube Device |
| CCTR | Continuum Concentric Tube Robot |
| CTCR | Concentric Tube Continuum Robot |
| – | Concentric Tube Manipulators |
| CTR | Concentric Tube Robot |
| – | Nested Cannulas |

TABLE II: Content of the constructed data set. The data set has 139 data points. Each data point contains five entries. Each entry consists of a key and a value.

| Key | Content of the value |
|---|---|
| authors | A list of authors with first and last name |
| robot | An unique ID identifying similar prototype within all paper in the data set. |
| year | Year of publication |
| cited_by | Number of citations according to GOOGLE SCHOLAR on the data of collection. |
| entrytype | Type of publication, i.e. journal article or conference paper. |

## A. Concentric Tube Continuum Robot

A CTCR is composed of multiple concentric tubes which are pre-curved and super-elastic and, therefore, inherently compliant and flexible. Inherited by the fact of being a continuum robot, the overall shape of the tube conforms a curve with continuous tangent vectors [5] characterized by the highly non-linear behaviour due to the elastic interaction between the tubes. In general, the flexible tubes are made of a metal alloy of nickel and titanium or made of thermoplastic materials [1, 39] in some instances. The material can be arbitrary as long as the elastic interaction between the tubes can be exploit. However, there is no precise definition of a CTCR regarding its number of tubes and degrees of freedom. It is crucial to define a CTCR in order to determine which paper and which robot should be taken into account and included in the data set. From this necessity, we define a CTCR as follows:

*Definition: Concentric tube continuum robots (CTCR) are composed of at least two concentric tubes, creating at least four degrees of freedom by translating and/or rotating each tube with respect to each other. The shape of the CTCR is an imaginary centre line formed by the nested tubes describing a smooth curve in space with a continuous tangent vector.*

## B. Data Acquisition

Using GOOGLE SCHOLAR, 139 scientific papers were gathered during the period of April/May 2020, including only peer-reviewed conference papers and journal articles relating to CTCR, based on our definition above. The *allintitle* operator of GOOGLE SCHOLAR was used with key words listed in Table I. Furthermore, potential entries of the data set are publication cited by authors of the first known publications [42, 48] and first surveys [14, 35]. We also consider potential entries citing the publications [14, 35, 42] and [48]. To increase quality of data set, the possible entries for the data set are reviewed and filtered afterwards according to its relevance to the research community and the proposed definition of a CTCR. The final data set consists of 139 data points, with each data point corresponding to one of the extracted publications. The entries for each data point are listed in Table II. Further, we count book chapters as journal articles.

## C. Leverage of Sharing

In order to quantify the effect of a prototype on the research community, one can count its usage in different publications. However, the counted number does not indicate how and if the prototype is distributed at least among the peers in the research group. For instance, a highly productive individual researcher can easily reuse a prototype, even though its usage is not shared among colleagues. Hence, a general statement about the impact of a particular prototype on the research community cannot be quantified by naively counting the publications. Therefore, we define a metric that takes into account the growing number of authors using the same prototype. The proposed metric, which we refer to as *Leverage of Sharing*, is quantified by

$$\mathcal{LS} = \sum_{i=1, \mathcal{A}_i \neq \mathcal{A}_j}^{n} \left( 1 - \frac{|\mathcal{A}_i|}{|\mathcal{A}|} \right) \quad \text{with } \mathcal{A} = \bigcup_{i=1}^{n} \mathcal{A}_i, \quad (1)$$

where $\mathcal{A}_i$ and $n$ are the set of authors in the $i^{\text{th}}$ paper and the number of papers using a particular prototype, respectively. The operator $|\cdot|$ acting on the set gives the number of individual authors. The notation $\mathcal{A}_i \neq \mathcal{A}_j$ under the summation indicates that only unique sets of authors are considered. Each summand in (1) computes the relative growth in usage with respect to the $i^{\text{th}}$ paper.

For the sake of clarification, we provide two toy examples, where (1) are applied. Consider three publications, the first two publications both have authors A and B, while the third publication has authors A and C, then we get $|\mathcal{A}_1| = |\mathcal{A}_2| = |\mathcal{A}_3| = 2$ and $|\mathcal{A}| = 3$. Applying by (1) results in $\mathcal{LS} = (1 - 2/3) + (1 - 2/3) = 2/3$ because of $\mathcal{A}_1 = \mathcal{A}_2 = \{A, B\}$. In the second toy example, two publications with $\mathcal{A}_1 = \{A, B\}$ and $\mathcal{A}_2 = \{C, D\}$ are considered. After the second publication, the number of authors increased by $100\%$ from the first publication, and we compute $\mathcal{LS} = 1$. Therefore, a higher $\mathcal{LS}$ relates to more collaboration and sharing.

## D. Causal Loop Diagrams – A Systems Thinking Tool

To conduct a meta-analysis, we apply causal loop diagrams, displaying the dynamics in the research community by indicating the causal relationships between key variables. In particular, two archetypes illustrated in Fig. 3 are briefly introduced and the notation listed in Table III are used. We refer to Kim [27, 28] for a gentle introduction.

*a) Shifting the Burden:* This describes the implication of solving a problem by applying a symptomatic solution instead of its fundamental solution. In general, symptomatic solutions are trivial and immediately implementable, leading to short-term relief of the problem. However, they create side

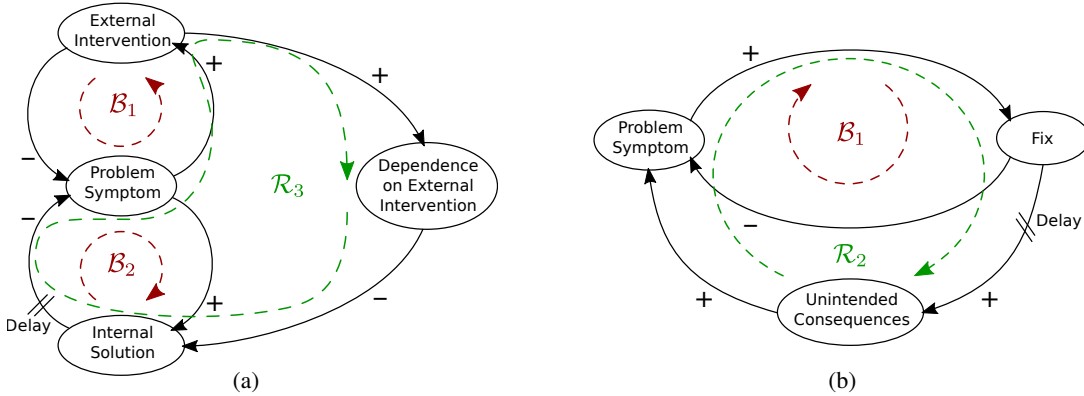

Fig. 3: Used systems archetypes at a glance. (a) *Shifting the Burden*. (b) *Fixes That Fail*. (Figures adapted from [27])

TABLE III: Notation used to describe feedback loops in causal loop diagrams.

| Symbol | Comment |
|---|---|
| $\rightarrow$ | A directed link between two variables. |
| // | Indicates that a directed link has a delay. |
| + | Indicates that a directed link has positive change, i.e. a change between adjoined variables has the same effect. |
| $-$ | Indicates that a directed link has negative change, i.e. a change between adjoined variables has opposite effect. |
| $\mathcal{R}$ | Describes a feedback loop as reinforcing. |
| $\mathcal{B}$ | Describes a feedback loop as balancing. |

effects in the long run and divert attention away from the fundamental problem. When the side effect solidifies and the problem symptom overwhelms, this archetype shows addiction patterns and becomes the *Addiction* archetype.

*b) Fixes That Fail:* Applying a fix to the problem without considering the full impact of this fix can lead to unintended consequences, as shown by the *Fixes That Fail* archetype. In the short-term, the problem is alleviated. However, unintended consequences form a reinforcing feedback loop that exacerbate the problem in the long term. In order to break the cycle presented by this archetype, Kim in [27] suggests acknowledging that the fix is only alleviating a symptom and, consequently, making a commitment to solve the fundamental problem.

## IV. ANALYSIS OF THE RESEARCH COMMUNITY

This section describes further motivations for researcher to use a variety of early-stage prototypes. It is an investigation of the dynamics between rate of publication, number of research groups, and number of different prototype. The resulting causal loop diagram depicted in Fig. 4 is mainly based on our subjective experience and perception regarding research community. Afterwards, the gathered data set is evaluated.

### A. Publication Pressure - A System Thinking View

A researcher is commonly measured by citation indices and number of publications. Therefore, an accepted publication can increase positive awareness of the researcher and the institution, leading to funding and career enhancement. Especially in the *tenure process*, where, researchers early in their careers face pressure to publish scientific papers in order to succeed. As a result, the phrase *publish-or-perish* was coined describing the phenomenon.

Another phenomenon is best described by a recommendation one might receive at the start of an academic career: *"Shoot for two paper submissions per year. On average you will get one accepted paper. And after four years you get the three to four papers for your Ph.D. thesis."* This recommendation is based on the preference to publish at a prestigious conference and its acceptance rate. Indicating the high-quality of an accepted paper, premier conferences like ICRA and IROS as well as RSS have an average acceptance rate of $45.6\%$, $42.1\%$, and $28.6\%$, respectively. The middle part of the recommendation may refer to advisor's experiences on the accuracy of conference reviews and acceptance process.

A more specific phenomenon concerning robotics is to aim for a so-called *strong paper*. Since evaluations in simulation are limited to the capability of the often over-simplified simulator, a robot-based evaluation of the proposed method is strongly recommended. This demand partially derives from reviewers during a review procedure. A paper including robot-based evaluation meaning an evaluation with a physical robot is considered to require more effort and to create more impact. Being in competition to be accepted for publication among other submissions, it might be desirable to aim for a *strong paper* or just to add a picture of a robot in the manuscript.

Now, we describe the influence of publication pressure on the usage and development of a robotic platform. The interaction between the components of the system are modeled with a causal loop diagram, shown in Fig. 4. The designed system has three main components. First, the publication pressure and the need for a robot-based evaluation. The researcher then has two options, the usage of a prototype or of a robotic platform. Therefore, the second main component consists of the usage of a prototype and its possible development. Analogously, the third main component consists of the development and usage of a robotic platform. The interactions between the components of the system are mainly described by two balancing loops, i.e. $\mathcal{B}_{\mathrm{pt}}$ and $\mathcal{B}_{\mathrm{ctcr}}$, and two reinforcing loops, i.e. $\mathcal{R}_{\mathrm{pt}}$ and $\mathcal{R}_{\mathrm{ctcr}}$. Both balancing loops describe the relief from publication pressure in the sense that a *strong paper* can be submitted which eventually has a higher chance of being accepted. This relief can be triggered by the use of a prototype described by $\mathcal{B}_{\mathrm{pt}}$ or by the use of a robotic platform described by $\mathcal{B}_{\mathrm{ctcr}}$. The reinforcing loop $\mathcal{R}_{\mathrm{pt}}$ reduces the chance

TABLE IV: Five prototypes found in the data set. These prototypes appears at least five times in the data set. The set of authors $\mathcal{A}$ and the metric $\mathcal{LS}$ are described in (1) in Sec. III-C. The average value for citation, number of authors $\mathcal{A}$, and $\mathcal{LS}$ is $101.6$, $7.3$, and $0.655$, respectively.

| years | $\mathcal{A}$ | $\mathcal{LS}$ | citations | publications |
|---|---|---|---|---|
| 2011 – 2016 | 23 | 5.870 | 475 | [6], [4], [3], [15], [29], [32], [46], [49] |
| 2010 – 2019 | 19 | 4.578 | 391 | [7], [13], [18], [19], [20], [26], [37], [38] |
| 2010 – 2016 | 14 | 2.643 | 426 | [2], [14], [31], [41], [40] |
| 2015 – 2018 | 12 | 2.167 | 54 | [30], [36], [43], [45], [44] |
| 2009 – 2011 | 6 | 2.333 | 780 | [10], [12], [11],[33], [34] |

of using a robotic platform by diverting more time and effort towards creating a prototype. While the publication pressure continues to reinforce the need for a robot-based evaluation, $\mathcal{R}_{\mathrm{pt}}$ demonstrates an incentive to use a prototype. Following the analogy, $\mathcal{R}_{\mathrm{ctcr}}$ describes an inclination to use a robotic platform. However, $\mathcal{R}_{\mathrm{ctcr}}$ has a crucial delay, as creating a robotic platform consumes more time and effort in comparison to creating a prototype. Therefore, $\mathcal{R}_{\mathrm{pt}}$ is faster than $\mathcal{R}_{\mathrm{ctcr}}$.

### B. Publication behaviour – A data-driven view

The gathered data set comprises data points associated with $139$ publications, where $84$ are published at conferences and $55$ appears in journals. Publication with robot-based evaluation account for $50\%$, $58.2\%$, and $53.2\%$ for conference paper, journal articles, and all publications, respectively. It can be found that the publication rate peaked around 2015 and that the number of new publications is slowly decreasing after 2015, with the rate of publications at conferences decreasing faster whereas the rate of journal articles are nearly constant. Figure 5 shows publication rate as well as the cumulative number of conference papers over time.

### C. Prototypes in the data set

We found 36 different prototypes. Ordering the prototypes by number of appearances in the data set reveals a Pareto distribution, where the first prototype appears eight times, the first 14 prototypes are used in 52 publications and the other 22 prototypes are each utilized only in one publication. Therefore, $61.1\%$ of all prototypes are used only once. Figure 6 shows how often the prototype appears at least a second time in a publication over time.

Applying the proposed *Leverage of Sharing* to the data set, we found a Pareto distribution where as the prototypes with the highest $\mathcal{LS}$ are listed in Table IV and only 12 prototypes have non-zero $\mathcal{LS}$ values. Thus, 24 prototypes are not shared with other peers and not even in the same research group.

## V. DISCUSSION

The data set and Fig. 5 confirm the widely held opinion in robotics research that strong papers often need to include robot-based experiments. Combined with the perceived pressure to publish as fast as possible, this often creates an immediate need for prototypes. To overcome this challenge, two possible solutions are presented in Fig. 4. First, building a simple prototype for the scope of the publication. Second, developing a robotic platform that might be shared with the research community to be reused by peers. However, considering the circumstances illustrated in Fig. 4, it is often far easier and more feasible to create a prototype specifically tailored to the purpose of the envisioned publications than to work on a reusable, organized robotic platform, as this would take more time and efforts. This habit can be verified by the data set and illustrated in Fig. 6.

The causal loop diagram in Fig. 4 is reminiscent of two archetypes, i.e. *Shifting the Burden* and *Fixes that Fail*. Following the *Fixes that Fail* archetype, developing a prototype addresses only the symptoms, i.e. the need for a robot-based evaluation, and not the fundamental problem, i.e. the lack of an accessible robotic platform. At the same time, the research community increasingly depends on prototypes, as it becomes harder to develop a robotic platform due to publication pressure, the limited time for an aspiring researcher has as well as the proven effectiveness of prototypes. Since robot-based evaluation using prototypes is an effective practice, indicated by Fig. 6, creating a prototype has a higher likelihood of succeeding than creating a robotic platform. The short-term yet quickly-reached success of the practice justifies promoting this practice. This shifts the attention from a robotic platform and diminishes its successful development, which is a further justification for allocating more resources to the developing of a prototype. This behavior is analogous to the *Shifting the Burden* archetype. The counterintuitive behaviors bolstered by publication pressure drives the existing lack of a robotic platform.

Only $38.9\%$ of the identified prototypes are used for more than only one publication, demonstrating the inefficient habit of not reusing existing prototypes. Since it takes time and effort to develop such prototypes, this habit limits the time available for more advanced research endeavors. However, once an ubiquitous robotic platform is available, this would minimize the repetitive tasks of developing a prototype, allowing researchers to focus on the main research challenges. In addition, robot-based evaluations can be performed quickly and are fully reproducible.

Finally, we acknowledge that it might be advisable to iterate through many prototype designs in order to investigate different approaches to find the most feasible design for a more mature robotic platform. Furthermore, new insights or newly developed technologies might require new and advanced prototypes to take advantage of them. That being said, it is likely that other factors among other such as monetary cost of developing, required engineering effort, and expected profits limit the emerge of a CTCR platform. However, startups like Virtuoso Surgical Inc. aimed to commercialize CTCR show that it is worthwhile to invest time, effort, and money in the development of a reliable CTCR platform. Nevertheless, an accessible CTCR platform could democratized and leveraged the community to achieve the necessary advancement and steps toward promising applications such as robot-assisted minimally invasive surgery and inspection.

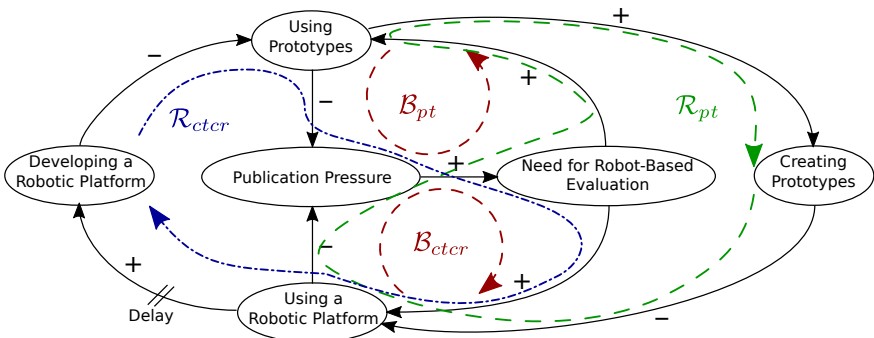

Fig. 4: Prototype development promoted by publication pressure in the research community. The delay in $\mathcal{R}_{ctcr}$ influences the dynamic in the causal loop diagram.

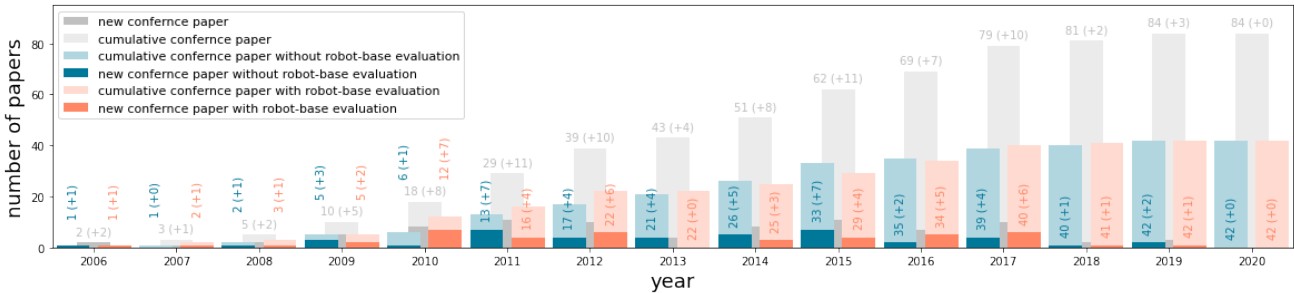

Fig. 5: Published conference papers over time. Six different groups of bars are shown, which can be divided into three categories and two chronological sequences. The first type of sequence, indicated with a darker color, describes the rate of publication. As a second type, a light color marks the cumulative number of all paper up to the respective year. The groups indicated by a blue or red color shows the publication includes a robot-based evaluation or not. The grey colored group does not distinguish between the them. Note that data points for the year 2020 are not complete due to the time of our data collection, see Sec. III-B.

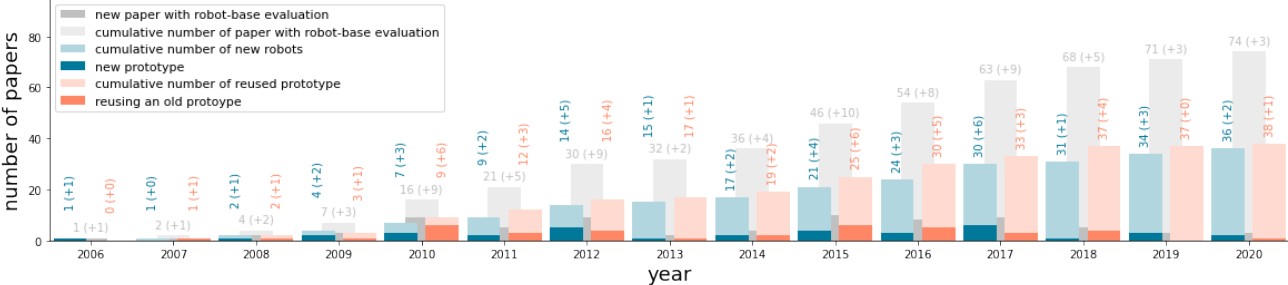

Fig. 6: Reusing and creating robot prototypes estimated by publications over time.

## VI. CONCLUSIONS AND FUTURE DIRECTIONS

In this work, we discuss the impact of the current habit of developing a variety of different prototypes on the emergence of a ubiquitous robotic platform. System thinking tools are utilized to model this interaction which is further verified using a manually gathered data set considering 139 publications. This is mainly caused by the omnipresent publication pressure and the resulting need for robot-based evaluations. Encouraged by the relatively inexpensive and simple construction of a prototype, researchers might follow the habit to develop an individual prototype for the sake of increasing the chance of an accepted publication. However, a more sustainable approach to this problem would be to strive for a common robotic platform which might become the sought-after *white rat* of CTCR research.

## ACKNOWLEDGMENTS

We acknowledge the support of the Natural Sciences and Engineering Research Council of Canada (NSERC).

Some of the results in this manuscript appeared in a term paper authored by the first three authors and prepared for the course called *Systems Thinking for Global Problems* taught by Steve Easterbrook at the University of Toronto. Therefore, we would like to thank him for discussing and sharing his knowledge on related topics as well as teaching us the tools that have greatly inspired the course of this research.

We would like to thank Thien-Dang Nguyen for the illustrations in Fig. 1. Further, we would like to thank Charlotte Tkany and Saskia Golz for the permission to use the image in Fig. 2d.

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
