# OpenReview forum: "CTCR Prototype Development: An Obstacle in the Research Community?"
_roboticsfoundation.org/RSS/2020/Workshop/RobRetro — RobRetro 2020_

### Official Review · AnonReviewer1 · 2020-06-20
**Retrospective on the dynamics between publication pressure and developing CTCR prototypes**

**Rating:** 8
**Confidence:** 3

**Review:**

This article presents a retrospective on the development of Concentric Tube Continuum Robotics (CTCR), and provides a data driven analysis of the current habits of the research community. This was done by defining a set of CTCR prototypes, collecting a dataset of 139 research papers that used these hardware prototypes and analysing them via a proposed metric called the “Leverage of Sharing” that quantified the relative growth in usage of a specific prototype. Further, a meta analysis based on causal feedback loops was performed which identifies the need for real-world evaluation and high frequency of publications as primary reasons for the prevalent behaviour of proposing novel prototypes as opposed to developing a standardised robotic system for research (the so called “white rat”).

Overall, the paper is nicely written and presents significant evidence in support of its claims. It is interesting to see the dynamic between publication pressure and the preference for short term solutions such as the focus on new prototypes. Many of the points raised such as the pressure on a researcher for publishing frequently, and expectations of reviewers in terms of real world evaluations etc. are quite general and extend to many areas of robotics. While these could be possible reasons for the over-reliance of the community on CTCR prototypes, it need not be the only factors that limit the development of a standardised robotic platform. It is possible that other factors such as the monetary cost of developing standardised platforms, time to get certification, expected profits from such a platform, engineering effort required etc. also provide significant blocks. While the authors cite manipulation platforms such as PUMA, DLR and PANDA, many of these were developed specifically to target the growing automation market where it was quite obvious what the monetary gain from such a platform would be — such monetary gain often can act as a strong motivator for industry-academic collaboration, thereby fostering faster research cycles. It would be nice if the authors can add a discussion on such broader market factors and their effective contribution towards the lack of standardised CTCR platforms.

---

### Decision · Program_Chairs · 2020-06-25

Accept